# Towards a proof of the Weak Gravity Conjecture

**Alfredo Urbano**[a]

[a] *INFN, sezione di Trieste, SISSA, via Bonomea 265, 34136 Trieste, Italy.*

**Abstract**

The Weak Gravity Conjecture (WGC) asserts a powerful consistency condition on gauge theories coupled to gravity, and it is widely believed that its proof will shed light on the quantum origin of gravitational interactions. Holography, and in particular the AdS/CFT correspondence, is a well-suited tool by means of which it is possible to explore the world of gravity. From the holographic perspective gravity is an emergent statistical phenomenon, and the laws of gravitation can be recast as the laws of thermodynamics. It is interesting to ask whether the WGC can be formulated in terms of the AdS/CFT correspondence. A positive answer is given: The WGC in the bulk is linked to the thermalization properties of the CFT living on the boundary. The latter is related to the Sachdev-Ye-Kitaev model of strange metals. In the thermodynamic picture, the validity of the WGC is verified.

# 1    Introduction

The Weak Gravity Conjecture (WGC) [1] imposes a powerful consistency condition on gauge theories coupled to gravity. In its original formulation (sometimes dubbed "electric" WGC), it can be stated as follows.

**W**eak **G**ravity **C**onjecture
*A $U(1)$ gauge symmetry coupled consistently to gravity requires the existence of at least one state with charge larger than its mass $\mu$ in Planck units*

$$qe > \frac{\mu}{M_\mathrm{P}} \tag{1}$$

where $e$ is the $U(1)$ gauge coupling and $q$ the charge of the state in units of $e$.[1] A rigorous mathematical proof of the WGC so far escaped the net. However, there are many pieces of evidence that make us think it is correct.

#1 *Trouble for remnants.* Consider a charged black hole with charge $Q$ and mass $M$ decaying into particles with mass $\mu$ and charge $qe$. Conservation of charge gives the number of particles in the final state, $N = Q/qe$. Conservation of energy implies $\mu N = \mu(Q/qe) < M$ from which it follows that a charged black hole decays if there exist lighter states with higher charge-to-mass ratio. For an extremal black hole with $Q = M/M_\mathrm{P}$, this argument implies $\mu/M_\mathrm{P} < qe$. Said equivalently, if the WGC fails extremal black holes are exactly stable remnants, a situation that seems to imply some pathologies [2, 3].

#2 *No global symmetries can exist in a theory of quantum gravity.* The WGC poses an obstruction to the limit of vanishing gauge couplings. This is in agreement with the conjecture according to which there are no global continuous symmetries in models of quantum gravity [4]. If the WGC fails, it will be possible to emulate a global symmetry by taking the limit $e \to 0$.

#3 *Infrared consistency.* A violation of the WGC could induce some pathologies in the infrared dynamics describing photons and gravitons [5].

#4 *Absence of explicit counterexamples.* The WGC was verified in a handful of explicit string theory constructions, in particular in simple heterotic setups [1] and in the framework of F-theory [6].

We remark that the WGC says nothing about the spin of the state in eq. (1). Furthermore, for the state satisfying eq. (1) we can say that "gravity is the weakest force" because the Coulomb-like repulsion (proportional to $q^2 e^2$) overcomes the gravitational attraction (proportional to $\mu^2/M_\mathrm{P}^2$).

---

[1]Despite the notation reminds that of electromagnetism, we have in mind a generic local $U(1)$ symmetry.

The WGC plays a central role in the so-called "Swampland program" that is the possibility to distinguish the Landscape (that is defined by the set of consistent low-energy effective field theories that are compatible with string theory) from the Swampland (that is defined by the set of consistent-looking low-energy effective field theories which are actually not compatible with string theory) using infrared data [7]. In this respect the WGC, if true, would provide, at least conceptually, a powerful discriminatory tool: Remarkably, by simply inspecting the low-energy spectrum, it might be possible to catalogue a given effective field theory in one or the other set.

There are many other interesting conjectures – related or complementary to the original WGC – that aim at populating the swampland with effective field theories that are consistent-looking at low-energy but pathological as a consequence of their violation. Needless to say, increasingly strong constraints on low-energy effective field theories require more elaborated conjectures that are, consequently, more difficult to justify on the basis of simple black hole arguments. Of particular relevance are some proposed extensions of the WGC (motivated by the fact that the WGC is not stable under dimensional reduction) which imply that there must be not just one charged particle of appropriate mass but a whole tower of increasingly heavy charged states fulfilling the WGC, $\mu_i < q_i e M_{\rm P}$ [8–10]. This (much) stronger version of the WGC turns out to be closely connected to the Swampland Distance Conjecture [11] and the Completeness Conjecture [12].

Besides theoretical implications, there are also intriguing phenomenological consequences. In [13], connections between the WGC and naturalness in theories with fundamental scalar were discussed. Even more ambitiously, in [14] it was conjectured that the WGC (to be more precise, its generalized version in terms of $p$-form gauge fields [1]) implies that non-supersymmetric anti-de Sitter (AdS) vacua supported by fluxes are unstable. If true, this extension of the WGC would imply interesting constraints on neutrino masses, the cosmological constant, and generic beyond the Standard Model physics [15].

All these arguments motivate the necessity to prove the WGC beyond the folklore theorems that justify it. Interesting attempts in this direction are related to black hole entropy [16–18], the Cosmic Censorship [19], the anti-de Sitter/Conformal Field Theory (AdS/CFT) factorization problem [20], unitarity and causality arguments [21], and the universal relaxation bound [22, 23]. The proposal in ref. [23] is particularly intriguing because it puts forward the possibility to link the WGC to thermalization properties of black holes. Inspired by this connection, in this paper we explore the WGC using the language of the AdS/CFT correspondence [24–26].

The bottom line of this work is the following. The AdS/CFT correspondence is a natural tool for analyzing thermodynamic aspects of gravity. From the AdS/CFT perspective, gravity is an emergent statistical phenomenon, and it is possible to reformulate the laws of gravitations in terms of the laws of thermodynamics. It is therefore interesting to ask *i)* whether the WGC admits an AdS/CFT interpretation and *ii)* if the dual picture sheds some light on the validity of the WGC. We shall give a positive answer to these questions in section 3 and section 4. Our starting point is the simple observation that argument #1 enumerated before makes use

of extremal charged black holes to motivate the WGC. It is, therefore, a good idea to start our discussion by giving a closer look at the geometry of these objects. This will be the topic of the next section.

## 2    Near-extremal Reissner-Nordström black holes

In General Relativity, a black hole with mass $M$ and charge $Q$ is known as the Reissner-Nordström (RN) black hole. In four space-time dimensions with coordinates $(t, r, \theta, \varphi)$, the metric is given by the line element[2]

$$ds^2 = -f(r)dt^2 + \frac{dr^2}{f(r)} + r^2 \left( d\theta^2 + \sin^2 \theta d\varphi^2 \right) , \qquad f(r) = \left( 1 - \frac{2M}{r} + \frac{Q^2}{r^2} \right) , \qquad (3)$$

where we used a geometrized unit system with $c = G_N = 1$. The RN spacetime has two horizons located at $r_\pm \equiv M \pm \sqrt{M^2 - Q^2}$, the outer at $r = r_+$ being the event horizon of the black hole.[3] The area of the event horizon is $A_{\rm RN} = 4\pi r_+^2$, and the Bekenstein-Hawking entropy $S_{\rm RN} = A_{\rm RN}/4 = \pi r_+^2$. The black hole temperature is

$$T_{\rm RN} = \frac{r_+ - r_-}{4\pi r_+^2} = \frac{\sqrt{M^2 - Q^2}}{2\pi \left[ M + \sqrt{M^2 - Q^2} \right]^2} . \qquad (4)$$

For a Schwarzschild black hole $T_{\rm Sch} = \lim_{Q \to 0} T_{\rm RN} = 1/8\pi M$. The expression of the temperature can be obtained by Wick rotating and compactifying the Euclidean time, and identifying the period with the inverse temperature. More physically, it is easy to show that the first law of thermodynamics

$$dM = TdS + \xi dQ , \qquad T = \left( \frac{\partial M}{\partial S} \right)_Q , \quad \xi = \left( \frac{\partial M}{\partial Q} \right)_S = \frac{Q}{r_+} , \qquad (5)$$

is satisfied for the above expressions of temperature and entropy. In eq. (5), $\xi$ is the electric potential and it plays the role of chemical potential. The gauge field for the black hole solution

---

[2] This is an exact solution of the Einstein-Maxwell theory in which the source term in the Einstein field equations is the energy-momentum tensor of the electromagnetic field of a point charge. Consider the action

$$\mathcal{S}_{\rm EM} = \frac{1}{16\pi G_N} \int d^4x \sqrt{-g} \mathcal{R} - \frac{1}{16\pi} \int d^4x \sqrt{-g} F_{\mu\nu} F^{\mu\nu} + \int d^4x \sqrt{-g} \mathcal{L}_{\rm M}[\Phi_i(x), \partial_\mu \Phi_i(x)] , \qquad (2)$$

for a $U(1)$ gauge theory coupled to gravity. $\mathcal{R}$ is the Ricci scalar and $\mathcal{L}_{\rm M}$ describes, in full generality, all matter fields $\Phi_i$ appearing in the theory with equations of motion $\delta \mathcal{L}_{\rm M}/\delta \Phi_i = 0$. The RN is a solution of Einstein field equations $\mathcal{R}_{\mu\nu} - \mathcal{R} g_{\mu\nu}/2 = 8\pi T_{\mu\nu}$ coupled to the Maxwell equations $\nabla_\mu F^{\mu\nu} = 0$, with $F_{\mu\nu} = \partial_\mu A_\nu - \partial_\nu A_\mu$. The source term is $T_{\mu\nu} = (g^{\rho\sigma} F_{\mu\rho} F_{\nu\sigma} - g_{\mu\nu} F_{\rho\sigma} F^{\rho\sigma}/4)/4\pi$ and there is no contribution from the matter fields, $T_{\mu\nu}^{\rm M} = -\frac{2}{\sqrt{-g}} \frac{\partial \sqrt{-g} \mathcal{L}_{\rm M}}{\partial g^{\mu\nu}} = 0$. Notice that in the geometric unit system the charge $Q$ has dimension $[Q] = [L]$ while it is dimensionless in natural units where the metric function takes the form $f(r) = 1 - 2G_N M/r + G_N Q^2/r^2$.

[3]The inner horizon is a Cauchy horizon that is the boundary of the region which contains closed time-like geodesics.

in eq. (3) takes the form $A_\mu dx^\mu = A(r)dt$ with[4]

$$A(r) = Q\left(-\frac{1}{r} + \frac{1}{r_+}\right) , \qquad A(r_+) = 0 . \tag{6}$$

The chemical potential of the black hole is the value of $A(r)$ at $r \to \infty$, and its expression matches that of $\xi$ in eq. (5). To avoid the presence of a naked singularity, the black hole charge must satisfy the bound $Q \leqslant M$. The extremal RN black hole has $Q = M$. In this limit, the two horizons $r_\pm$ coincide. This is an interesting limit since an extremal RN black hole behaves like a thermodynamic system with ground state degeneracy: It has finite entropy $S_{\mathrm{RN}}^{\mathrm{ext}} = \pi M^2$ with vanishing temperature $T_{\mathrm{RN}}^{\mathrm{ext}} = 0$. Another quantity of interest is the behavior of the heat capacity as a function of the black hole charge. We find (cf. fig. 1)

$$C_{\mathrm{RN}} = T\left(\frac{\partial S}{\partial T}\right)_Q = \frac{2\pi r_+^2 \sqrt{M^2 - Q^2}}{M - 2\sqrt{M^2 - Q^2}} . \tag{7}$$

Far from extremality, the heat capacity is negative for $Q < \sqrt{3}M/2$. This property can seem paradoxical at first – a RN black hole with $Q < \sqrt{3}M/2$ gets hotter as it radiates energy – but it turns out to be rather ordinary for black holes with asymptotically flat spacetimes (including the simplest case of the Schwarzschild black hole for which $C_{\mathrm{Sch}} = -8\pi M^2$) as well as for some gravitationally bound systems that do not meet the strict definition of thermodynamic equilibrium, such as stars. This is not, indeed, the interesting part of the story. What is most surprising is that for a RN black hole the specific heat diverges at $Q = \sqrt{3}M/2$ and becomes positive for $Q > \sqrt{3}M/2$. Going towards the extremal limit, therefore, some sort of phase transition takes place and the black hole turns from a thermodynamic oddity into a more ordinary object. A RN black hole with positive heat capacity is to all effects a well-behaved thermodynamic system since it can be in equilibrium with a surrounding heat bath.

These arguments motivate a more quantitative analysis of the extremal limit. Let us start considering the extremal case with $Q = M$. We define the new variables

$$r = Q\left(1 + \frac{\lambda}{z}\right) , \qquad t = \frac{Q\tau}{\lambda} , \tag{8}$$

where $\lambda$ is an arbitrary small parameter. The position of the horizon corresponds to $z \to \infty$. From eq. (3), by taking the limit $\lambda \to 0$, we find the *near-horizon metric of the extremal RN black hole*

$$ds^2 = \underbrace{\frac{Q^2}{z^2}\left(-d\tau^2 + dz^2\right)}_{AdS_2 \text{ (Poincare patch)}} + \underbrace{Q^2\left(d\theta^2 + \sin^2\theta d\varphi^2\right)}_{S^2} , \qquad A_\tau(z) = \frac{Q}{z} \tag{9}$$

A new geometry has emerged [27]. The extremal limit is a limit of enhanced symmetry since the metric in eq. (9) is $AdS_2 \times S^2$. The extremal RN background in the near-horizon limit

---

[4]In geometric units the function $A(r)$ is dimensionless.

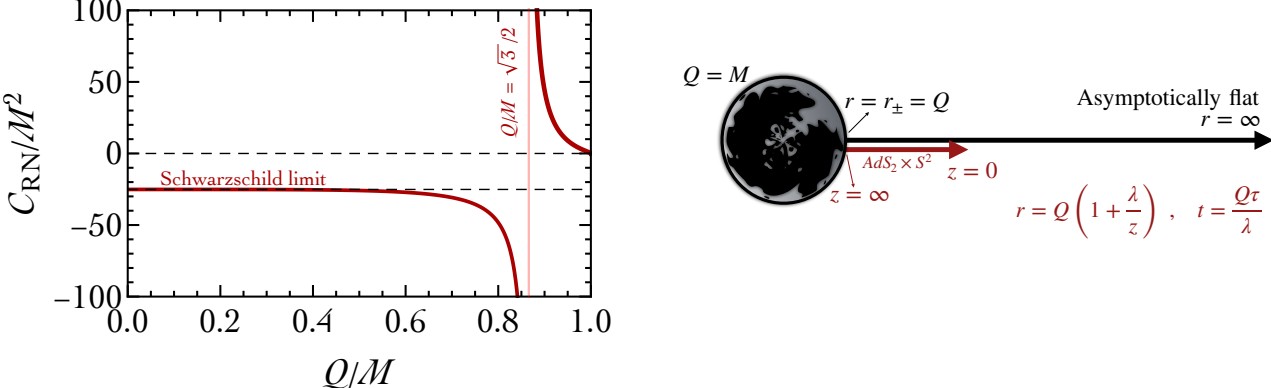

Figure 1: *Left panel. Heat capacity for a RN black hole, eq. (7). Close to extremality ($Q/M = 1$), the heat capacity becomes positive. Right panel. Sketch of the near-horizon geometry (red) in eq. (9).*

enjoys an $SO(3)$ isometry acting on the sphere $S^2$. This is nothing but the spherical symmetry of the original black hole solution. In addition, the metric also has an $SO(2,1)$ isometry acting on the $AdS_2$ space that was not present in the original solution. For an extremal RN black hole, in the near-horizon limit an effective two-dimensional AdS geometry is realised in the time-radial direction.[5] Notice that for an extremal black hole there is only one scale, the charge $Q$ corresponding to both the radius of the sphere $S^2$ and the radius of the $AdS_2$ factor.

The presence of the $AdS_2$ factor in eq. (9) suggests the possible existence of an AdS/CFT correspondence. Technically speaking, the key point is that the boundary of the near-horizon geometry, located at $z \to 0$, inherits the full conformal group from the isometries of $AdS_2$. We are facing a special case of the AdS/CFT correspondence with a one-dimensional conformal field theory (a.k.a. conformal quantum mechanics): The boosts and rotation in $AdS_2$ close, in the limit $z \to 0$, the one-dimensional conformal algebra[6] generated by dilatations, translations in $\tau$, and special conformal transformations (cf. [28]). Notice that the limit $z \to 0$ does not cover the full range of the original radial coordinate (sketch in the right panel of fig. 1).

The above construction can be generalized to the more interesting case of finite temperature. In addition to the variables defined in eq. (8), we shift the position of the horizon from

---

[5]Taking the near-horizon limit of the black hole metric is geometrically equivalent to a low-energy limit. This can be understood by noting that the redshift factor $f(r)$ in eq. (3) is non-constant as a function of $r$. As a consequence the energy $E$ of a particle measured by an observer at constant radial position $r$ differs from the energy of the same object measured by an observer at infinity, $E_\infty = E\sqrt{f(r)}$. In the near-horizon region, $f(r)$ is strictly zero as $\lambda \to 0$, and from the point of view of an observer at infinity everything in the near-horizon region is infinitely redshifted. Said differently, the redefinition of the time variable in eq. (8) implies that finite values of $\tau$ correspond to the long-time limit of the original coordinate $t$. Thinking in terms of conjugate variables, therefore, the near horizon geometry only applies to the limit of low frequencies.

[6]In $D$ dimensions, the conformal group has $(D+1)(D+2)/2$ generators.

the extremal limit $r_+ = Q$ to

$$r_+ = Q \left(1 + \frac{\lambda}{z_0}\right) \ .$$
(10)

We obtain the *near-extremal near-horizon metric of the RN black hole*

$$ds^2 = \underbrace{\frac{Q^2}{z^2} \left[-\left(1 - \frac{z^2}{z_0^2}\right) d\tau^2 + \left(1 - \frac{z^2}{z_0^2}\right)^{-1} dz^2\right]}_{\text{finite temperature } AdS_2 \text{ factor (Rindler patch)}} + Q^2 d\Omega_2^2 \ , \qquad A_\tau(z) = \frac{Q}{z}\left(1 - \frac{z}{z_0}\right)$$
(11)

We remark that both eq.s (9, 11) are actual solutions of the Einstein-Maxwell field equations, as verified by explicit computation. The temperature associated to the metric in eq. (11) is $T = 1/2\pi z_0$. Notice that this is a dimensionless quantity since it refers to the variable $\tau$ which is defined in units of $Q$. In eq. (11) the coordinate $z$ ranges from $z = z_0$ at the horizon to $z = 0$ at the CFT boundary. The non-zero value of the temperature acts as a cut-off for the limit $z \to \infty$, and for $z \ll z_0$ the metric is essentially $AdS_2 \times S^2$.

We are interested in the near-extremal limit of the black hole – that is at temperatures $T \ll 1$ – and we shall study its response at low energies, $\omega \ll 1$ ($T \ll 1/Q$ and $\omega \ll 1/Q$ if we restore the $Q$-dimension of the $\tau$ variable)

$$T \ll \frac{1}{Q} \ , \qquad \omega \ll \frac{1}{Q} \ , \qquad T \sim \omega$$
(12)

Once the background metric is defined, a natural question is to determine its response to small perturbations. Isolated black holes in equilibrium are indeed idealized objects. For instance, during the first moments after its formation due to gravitational collapse of matter, a newly born black hole is in a perturbed state. Generally, black holes always have complex distributions of matter around them such as accretion disks, and they actively interact with their surroundings. These simple considerations motivate the study of black hole perturbations. Of course, it is hard to expect that electrically charged black holes play any fundamental role in observational astrophysics due to charge neutrality of the Universe. However, we remind that we are considering a generic $U(1)$ local symmetry, a situation in which it is possible to envisage phenomenologically viable scenarios where charged black holes – even close to extremality – are allowed [29].

Apart from these phenomenological motivations, studying the stability of solutions of Einstein field equations is essential to determine their validity. Clearly, all the arguments that support the WGC listed in section 1 assume that RN black holes are valid solutions all the way up to the extremal limit. In our analysis, we shall, therefore, impose stability of the RN metric against perturbations. In particular, we shall work under the assumption that the near-extremal near-horizon metric of the RN black hole is stable.

Let us first set the problem in general fashion. We consider the metric $g_{\mu\nu}$ and matter fields $\Phi_i$ as a sum of the unperturbed background values and the actual perturbations, $g_{\mu\nu} = g_{\mu\nu}^0 + \delta g_{\mu\nu}$ and $\Phi_i = \Phi_i^0 + \delta\Phi_i$. Solving the perturbation dynamics is a highly non-linear

problem since in Einstein field equations variations in the energy-momentum tensor imply an alteration of space-time geometry, which in turn involves a matter redistribution. However, assuming small perturbations, we can neglect terms of order $O(\delta g_{\mu\nu}^2)$, $O(\delta \Phi_i^2)$, $O(\delta g_{\mu\nu} \delta \Phi_i)$ or higher. Under this assumption one finds, using the Einstein field equations and the equations of motion for the matter fields, a set of linear equations for the perturbations $\delta g_{\mu\nu}$ and $\delta \Phi_i$. Let us now focus on the case of RN black holes. The RN background solution is not sourced by any matter fields (i.e. $\Phi_i^0 = 0$, cf. discussion below eq. (2)), and the field perturbations are not coupled to the perturbations of the metric. The former are, therefore, equivalent to the dynamics of test fields in the black hole background. At the linear level perturbations of the RN space-time can be performed in two ways: by studying the dynamics of test fields in the black hole background or by perturbing the black hole metric itself.

In the next section we shall study the dynamics of test fields in the RN background.

# 3 Emergent CFT and the WGC

We are in the position to study the propagation of generic perturbations in the background geometries in eq.s (9, 11). For illustrative purposes, we start from the near-horizon metric of the extremal RN black hole, and we focus on a charged scalar $\Phi$ (which is, therefore, necessarily complex) with charge $q$ in units of $e$ and mass $\mu$. The action for $\Phi$ is

$$\mathcal{S} = -\int d^4x \sqrt{-g} \left[ (D^\mu \Phi)^* (D_\mu \Phi) + \mu^2 \Phi^* \Phi \right] , \tag{13}$$

where $D^\mu \Phi = (\partial^\mu - igA^\mu)\Phi$, with $g \equiv qe$.[7] The equations of motion is given by

$$\left[ (\nabla^\nu - igA^\nu)(\nabla_\nu - igA_\nu) - \mu^2 \right] \Phi = 0 , \tag{14}$$

where the covariant derivative acts on a generic vector according to $\nabla_\mu v_\nu = \partial_\mu v_\nu - \Gamma^\alpha_{\mu\nu} v_\alpha$. The metric allows for the separation of variables, and we use the ansatz

$$\Phi(\tau, z, \theta, \varphi) = e^{-i\omega\tau} \sum_{l,m} e^{im\varphi} \phi_l(z) S_l(\theta) . \tag{15}$$

The angular part $S(\theta)$ solves the eigenvalue equation for the associated Legendre polynomials

$$\left[ \sin\theta \frac{d}{d\theta} \left( \sin\theta \frac{d}{d\theta} \right) + l(l+1)\sin^2\theta - m^2 \right] S_l(\theta) = 0 , \tag{16}$$

and regularity at the poles $\theta = 0, \pi$ imposes the usual quantization conditions on the azimuthal $l$ and magnetic $m$ quantum numbers $l = 0, 1, \ldots$ with $-l \leqslant m \leqslant +l$. The function $\phi_l(z)$ solves the differential equation

$$-\frac{d^2\phi_l}{dz^2} + \left\{ \frac{1}{z^2} \left[ \mu^2 Q^2 + l(l+1) \right] - \left( \omega + \frac{gQ}{z} \right)^2 \right\} \phi_l = 0 . \tag{17}$$

---

[7]In our convention for the geometric units $[\mu] = [g] = [1/L]$. The combinations $\mu Q$ and $gQ$ are, therefore, dimensionless.

This equation admits an exact analytical solution in terms of the Whittaker functions

$$\phi_l(z,\omega) = C_{\text{out}}\mathcal{W}_{-igQ,\nu_l}(2i\omega z) + C_{\text{in}}\mathcal{W}_{igQ,\nu_l}(-2i\omega z) , \tag{18}$$

$$\nu_l \equiv \sqrt{\frac{1}{4} + Q^2(\mu^2 - g^2) + l(l+1)} . \tag{19}$$

Physical solutions are characterized by ingoing boundary condition at the horizon. The latter is located at $z \to \infty$, see eq. (8). The Whittaker functions feature the asymptotic behavior $\mathcal{W}_{\kappa,\nu}(\zeta) \overset{\zeta\to\infty}{\approx} e^{-\zeta/2}\zeta^\kappa$. Consequently, the only solution in eq. (18) that is ingoing at the horizon is $\phi_l(z,\omega) = C_{\text{in}}\mathcal{W}_{igQ,\nu_l}(-2i\omega z)$. The crucial aspect of the computation is the behavior of the function $\phi_l(z)$ at the boundary of the near-horizon geometry, $z \to 0$. From the properties of the Whittaker functions we have

$$\mathcal{W}_{\kappa,\nu}(\zeta) \overset{\zeta\to 0}{\approx} \frac{\Gamma(2\nu)}{\Gamma(1/2 + \nu - \kappa)}\zeta^{1/2-\nu} + \frac{\Gamma(-2\nu)}{\Gamma(1/2 - \nu - \kappa)}\zeta^{1/2+\nu} . \tag{20}$$

We, therefore, find

$$\phi_l(z,\omega) \overset{z\to 0}{\approx} C_{\text{in}}\left[\frac{\Gamma(2\nu_l)(-2i\omega)^{1/2-\nu_l}}{\Gamma(1/2 + \nu_l - igQ)} z^{1/2-\nu_l} + \frac{\Gamma(-2\nu_l)(-2i\omega)^{1/2+\nu_l}}{\Gamma(1/2 - \nu_l - igQ)} z^{1/2+\nu_l}\right]$$

$$\equiv \mathcal{A}_l(\omega)z^{1-\Delta_l} + \mathcal{B}_l(\omega)z^{\Delta_l} , \tag{21}$$

with $\Delta_l$ defined as

$$\Delta_l \equiv \frac{1}{2} + \underbrace{\sqrt{\frac{1}{4} + Q^2\left\{\mu^2\left[1 + \frac{l(l+1)}{\mu^2 Q^2}\right] - g^2\right\}}}_{\equiv\ \nu_l \text{ in eq. (19)}} . \tag{22}$$

Notice that the expansion in eq. (15) is equivalent to a Kaluza-Klein (KK) reduction from $AdS_2 \times S^2$ to $AdS_2$. The hallmark of any KK reduction on a compact manifold is the appearance of an infinite but discrete tower of modes of increasing mass. In our case the field $\Phi$ propagating in $AdS_2 \times S^2$ gives rise to a tower of fields $\phi_l(z)$ in $AdS_2$ with KK spectrum

$$M_{\text{KK}}^{(l)} = \mu^2 + \frac{l(l+1)}{Q^2} , \quad l = 0, 1, \ldots . \tag{23}$$

The lightest mode with $l = 0$ has mass $\mu$. The non-zero KK modes start at $\sim 1/Q$, where the radius $Q$ of the sphere $S^2$ plays the role of compactification scale. As in any other extra-dimensional scenario, in the low energy effective theory below the energy $1/Q$ the KK modes can be neglected. We shall return on this point later.

According to the AdS/CFT dictionary there exists a correspondence between the scalar field $\phi_l$ propagating in the AdS bulk and the scalar operator $\mathcal{O}_l$ belonging to the CFT at the boundary. The conformal dimension of the scalar operator $\mathcal{O}_l$ is given by $\Delta_l$. The scalar operator $\mathcal{O}_l$ is dubbed *irrelevant* if $\Delta_l > 1$, *relevant* if $\Delta_l < 1$ and *marginal* if $\Delta_l = 1$. By imposing the condition that the conformal dimension stays real we find the Breitenlohner-Freedman bound

$$Q^2\left(\mu^2 - q^2\right) + \left(l + \frac{1}{2}\right)^2 > 0 . \tag{24}$$

Below the Breitenlohner-Freedman bound the AdS/CFT construction is unstable. On the gravity side, the instability is related to the Schwinger effect caused by the spontaneous production of charged particle/anti-particle pairs in the RN spacetime [30]. On the CFT side, the instability is related to causality violation. We will not contemplate this possibility any further in this section but it is important to remark that the presence of the instability is crucially related to the fact that we are dealing with a scalar perturbation. Bose-Einstein statistics plays indeed an important role. On the gravity side, the number of boson pairs produced spontaneously in a given state is not limited by statistics, while such limitation does exist for fermions due to Pauli blocking [31, 32]. This physical argument is the same that prohibits the existence of superradiance for fermions [33]. On the CFT side, causality violation manifests itself for scalars but not for spinors (e.g., cf. [34] for the $AdS_4/CFT_3$ case).

From eq. (21) it is evident that the term proportional to $\mathcal{A}_l(\omega)$ diverges as $z \to 0$ while the term $\mathcal{B}_l(\omega)$ is regular.[8] The former (latter) is called leading (sub-leading) term. According to the rules of the AdS/CFT correspondence the CFT two-point correlation function for the scalar operator $\mathcal{O}_l$ can be expressed in terms of the ratio of the coefficients of the sub-leading and leading asymptotes of the corresponding AdS massive scalar wave in eq. (21)

$$\langle \mathcal{O}_l(-\omega)\mathcal{O}_l(\omega) \rangle = \frac{\mathcal{B}_l(\omega)}{\mathcal{A}_l(\omega)} \ . \tag{25}$$

More formally, it can be shown that the leading behavior of the solution $\phi_l(z, \omega)$ at the boundary acts as a local source $J$ for the operator $\mathcal{O}_l$, and the sub-leading term corresponds to the expectation value $\langle \mathcal{O}_l \rangle_J$ in the presence of the source. The ratio in eq. (25) thus describes the reaction of the field to a source, and can be identified – up to an overall constant which is independent of $\omega$ – with the retarded Green's function $\mathcal{G}_R^{(l)}(\omega) = \mathcal{B}_l(\omega)/\mathcal{A}_l(\omega)$.

In our discussion the poles of the retarded Green's function play a central role. They are defined by the values of $\omega$ such that $\mathcal{A}_l(\omega) = 0$. According to eq. (21) and to the previous discussion, the condition $\mathcal{A}_l(\omega) = 0$ corresponds to the absence of a source term at the boundary $z \to 0$. On the gravitational side, this setup has a neat physical interpretation. Solutions of eq. (14) with ingoing boundary condition at the horizon and outgoing boundary condition at infinity are characterized by a discrete set of eigenfrequencies that are called quasi-normal modes. Quasi-normal modes describe the dissipative properties of a black hole, and for such reason they are defined by the absence of inward-directed waves generated by some source at spatial infinity (the presence of a non-zero source at infinity would keep the system perturbed, contrary the very same definition of dissipative dynamics). The eigenfrequencies corresponding to the quasi-normal modes have both a real and an imaginary part. The latter is negative, and gives the inverse of the relaxation time $\tau_d$ of the corresponding

---

[8] In the AdS background given by eq. (9), the boundary contributions to the norm of $\phi$ are $\lim_{z \to 0}[\mathcal{A}_l^2 z^{1-2\Delta_l} + \mathcal{B}_l^2 z^{2\Delta_l - 1} + \mathcal{A}_l \mathcal{B}_l]$, with an extra factor $1/z$ coming from the determinant of the induced AdS metric on the boundary. If $1 - 2\Delta_l < 0 \Rightarrow \Delta_l > 1/2$, as guaranteed by the Breitenlohner-Freedman bound, the term proportional to $\mathcal{A}_l^2$ always diverges in the limit $z \to 0$ while the term proportional to $\mathcal{B}_l^2$ always vanishes. For this reason the leading term $\mathcal{A}_l$ (sub-leading term $\mathcal{B}_l$) in eq. (21) is also called non-normalisable (normalisable) solution.

mode, $\omega = \mathbb{Re}[\omega] + i\mathbb{Im}[\omega] \equiv \mathbb{Re}[\omega] - i2\pi/\tau_d$. From the $\tau$-dependence of the solution, $e^{-i\omega\tau}$, a negative imaginary part describes an exponential damping with characteristic time-scale set by $\tau_d$. The quasi-normal modes control the black hole ringdown, that is the decay of the perturbed black hole toward its hairless state. This is a thermalization process: The relaxation towards thermal equilibrium after a perturbation. The quasi-normal mode with the smallest value of $-\mathbb{Im}[\omega]$ (equivalently, with the largest value of $\tau_d$) is the less damped mode, and it dominates the thermalization time-scale.

The ringdown of the black hole is dual to the thermalization process of the corresponding CFT [35]. In the CFT, the quasi-normal modes manifest themselves as Ruelle resonances, that appear as poles in the Fourier transform of the retarded Green's function. The interpretation of black hole quasi-normal modes as poles of the retarded Green's function in the dual CFT was first pointed out for BTZ black holes in [36].

From eq. (21) we find

$$\mathcal{G}_R^{(l)}(\omega) = e^{-i\pi\nu_l} \frac{\Gamma(-2\nu_l)\Gamma(1/2 + \nu_l - igQ)}{\Gamma(2\nu_l)\Gamma(1/2 - \nu_l - igQ)} (2\omega)^{2\nu_l} . \qquad (26)$$

We note that in this case there are no poles in the retarded Green's function. This means that there are no quasi-normal modes for an extremal RN black hole associated with the dynamics of a test charged scalar field. This is not surprising since this is the limit in which the black hole temperature vanishes. In order to have some interesting dynamics, we shall now move to consider the near-extremal near-horizon metric of the RN black hole with non-vanishing temperature $T = 1/2\pi z_0$. We have to solve eq. (14) in the background given by eq. (11). It is again possible to exploit the separation of variables, and for the angular part we find again the associated Legendre polynomials in eq. (16). The equation for $\phi(z)$ is given by

$$\frac{d^2\phi}{dz^2} + \left(\frac{2z}{z^2 - z_0^2}\right)\frac{d\phi}{dz} + \left\{-\frac{[Q^2\mu^2 + l(l+1)]}{z^2 - z^4/z_0^2} + \frac{\left[\omega + gQ\left(\frac{1}{z} - \frac{1}{z_0}\right)\right]^2}{1 - 2z^2/z_0^2 + z^4/z_0^4}\right\}\phi = 0 . \qquad (27)$$

This is a second-order ordinary differential equation with three regular singular points at $z = 0, z_0, \infty$ and, as such, it can be transformed into the hypergeometric differential equation. We find the two independent solutions

$$\phi_l(z, \omega) \sim \left(\frac{1}{z} - \frac{1}{z_0}\right)^{-\frac{1}{2} \mp \nu_l} \left(\frac{z_0 + z}{z_0 - z}\right)^{\frac{i\omega z_0}{2} - igQ} \times \qquad (28)$$

$$_2F_1\left(\frac{1}{2} \pm \nu_l + i\omega z_0 - igQ, \frac{1}{2} \pm \nu_l - igQ, 1 \pm 2\nu_l; \frac{2z}{z - z_0}\right) .$$

The upper sign gives the ingoing solution at the horizon. As done before, we have to consider the behavior of the solution at the AdS boundary in order to extract the retarded Green's function as the the quotient of the sub-leading to leading term. Using the properties of the hypergeometric functions, we find

$$\mathcal{G}_R^{(l)}(\omega) = (4\pi T)^{2\nu_l} \frac{\Gamma(-2\nu_l)\Gamma(1/2 + \nu_l - i\omega/2\pi T + igQ)\Gamma(1/2 + \nu_l - igQ)}{\Gamma(2\nu_l)\Gamma(1/2 - \nu_l - i\omega/2\pi T + igQ)\Gamma(1/2 - \nu_l - igQ)} , \qquad (29)$$

where we used the explicit definition of $T$ instead of $z_0$. The quasi-normal frequencies are, therefore, dictated by the poles of the Gamma function, and for the imaginary part we find

$$\mathbb{Im}[\omega_{n,l}] = -2\pi T \left( \frac{1}{2} + n + \nu_l \right)_{n=0,1,\dots} \quad , \qquad \nu_l = \sqrt{\frac{1}{4} + Q^2 \left\{ \mu^2 \left[ 1 + \frac{l(l+1)}{\mu^2 Q^2} \right] - g^2 \right\}} \quad (30)$$

We note that this result agrees with the expression for the quasi-normal modes obtained in [37]. In [37] the quasi-normal modes were computed considering the full metric outside the black hole horizon instead of limiting the computation to the near-horizon geometry as done in this paper. The agreement between the two results is due to the fact that in the low-energy limit in eq. (12) only the near-horizon geometry is relevant. The thermalization time-scale is set by the less damped mode with $n = 0$, and we find (remember our definition $\tau_d \equiv -2\pi/\mathbb{Im}[\omega]$)

$$\tau_d^{(n,l)} = \frac{1}{T\left(1/2 + n + \nu_l\right)} \quad \overset{n=0}{\Rightarrow} \quad \tau_d^{(0,l)} = \frac{1}{T\left(1/2 + \nu_l\right)} \; . \quad (31)$$

We are interested in the response of the system at low frequencies, $\omega \ll 1/Q$ (cf. eq. (12)). As discussed below eq. (23), in this limit the KK modes do not participate to the low-energy dynamics and the thermalization time-scale is

$$\tau_d \equiv \tau_d^{(0,0)} = \frac{1}{T\left(1/2 + \nu_0\right)} \; , \qquad \nu_0 = \sqrt{\frac{1}{4} + Q^2 \left( \mu^2 - g^2 \right)} \; . \quad (32)$$

We are now in the position to discuss the connection with the WGC. Notice that $\tau_d > 0$. Negative $\tau_d$ would correspond to an exponentially-growing unstable fundamental mode that would threat the stability of the black hole geometry. Most importantly, this would clash against the thermodynamic interpretation of quasi-normal modes according to which $\tau_d$ must be positive. From this perspective, it is totally reasonable to expect not just $\tau_d > 0$ but the actual existence of a lower bound on $\tau_d$ since the thermalization process cannot be arbitrarely fast. An educated guess is $\tau_d \gtrsim 1/T$.[9] This intuition is supported by many explicit examples. On the gravity side, in [35] the relation $\tau_d \gtrsim 1/T$ was obtained for scalar perturbations of Schwarzschild-AdS black holes in four, five, and seven dimensions. Furthermore, on the CFT side, there exists a lower bound on $\tau_d$ as $T \to 0$ in all many-body quantum systems that admit an AdS/CFT description, $\tau_d > c/T$ where $c$ is a temperature-independent positive constant. A remarkable example that is close to the construction proposed in this paper is the Sachdev-Ye-Kitaev (SYK) model [38–40], a maximally chaotic model of strongly-interacting Majorana fermions. In [41] a numerical study found a thermalization rate consistent with the expectation $\tau_d > c/T$. We shall discuss in more detail the connection with the SYK model in section 5.

---

[9]There are two possibilities, based on dimensional analysis: $\tau_d \sim M$ and $\tau_d \sim 1/T$. For a Schwarzschild black hole, $T_{\text{Sch}} = 1/8\pi M$ and the two scales are related. This is not true for a RN black hole close to the extremal limit where $M \sim Q \ll 1/T$, and we expect the thermalization time-scale to be dominated by the scaling $\tau_d \sim 1/T$.

All in all, we impose the condition $\nu_0 < 1/2$ to ensure the condition $\tau_d > 1/T$ on the thermalization time-scale and, after restoring units of $M_\text{P}$, we find

$$qe > \frac{\mu}{M_\text{P}} \tag{33}$$

that is precisely the inequality envisaged by the WGC.

The crucial question is: Does the result derived in this section represent a proof of the WGC? To answer this question, we remind once again that the WGC asserts that there must exist at least one state with $qe > \mu/M_\text{P}$. This means that any attempt for its proof must involve some robust argument that makes mandatory the existence of a state satisfying eq. (33). At first sight, our result does not imply *per se* the existence of such state – at face value it says that a charged scalar field appearing in the action in eq. (2) must satisfy eq. (33) in order to be consistent with the laws of black hole thermodynamics but it does not guarantee the presence of such charged scalar in the spectrum.

However, there is more. In our case what makes mandatory the existence of a state with $\mu < qeM_\text{P}$ is the condition $\tau_d > c/T$ on the thermalization time-scale. In the black hole literature, this is known as the universal relaxation bound [22,23,42]. As we discussed before, the universal relaxation bound can be beautifully related – in the spirit of the AdS/CFT correspondence – to the thermalization properties of the CFT living on the boundary of the black hole near-horizon geometry. Let us elaborate more on this crucial point, along the lines of [22]. As we discussed at the end of section 2, there are two kinds of perturbations, decoupled at the linear level: those that involve perturbations of the background metric (including the electromagnetic field in the case of a RN black hole) and those related to test fields propagating in the black hole background. In principle the two sets of perturbations have different thermalization time-scales, and at least one of the two must satisfy the condition $\tau_d > c/T$. Background metric perturbations are characterized by $\tau_d^\text{BG} \sim M$. In the case of a Schwarzschild black hole, we have $T = 1/8\pi M$ and we can conclude that these perturbations satisfy the scaling $\tau_d^\text{BG} \sim 1/T$. However, the relation $T = 1/8\pi M$ is only true for ordinary black holes with negative heat capacity (cf. section 2) since they get hotter and hotter as they radiate (that is for decreasing mass $M$). On the contrary, in the near-extremal limit RN black holes do not behave this way since their heat capacity becomes positive, and the relation $M \sim 1/T$ breaks down. Background metric perturbations still behave according to $\tau_d^\text{BG} \sim M \sim Q$ [43] but now $\tau_d^\text{BG} \sim Q \ll 1/T$ as $T \to 0$, and, therefore, these perturbations decay too fast to satisfy the condition $\tau_d > c/T$.

With background metric perturbations out of the way, we conclude that for a near-extremal RN black hole there must exist at least one particle whose perturbations set the thermalization rate $\tau_d > c/T$. For such state, our computation shows that eq. (33) must be satisfied. This concludes our attempt to demonstrate the WGC.

The fact that we derived our result for the specific case of a charged scalar field may raise some eyebrows (the WGC, according to its original formulation, does not seem to prefer any specific spin). In the next section, we shall discuss the case of spin-1/2 particles.

# 4   What about the spin?

As mentioned in section 1, the WGC says nothing about the spin of the state whose mass and charge are involved in eq. (1). This could indicate that whatever physical argument is supporting the WGC it should be independent from the spin. In section 3 we derived the WGC considering the specific case of a scalar particle. It is interesting to investigate higher-spin dynamics. If the thermodynamic interpretation of the WGC proposed in this paper is correct, we expect to obtain the same result considering particles with different spin, and in this section we shall discuss the case of a spin-1/2 particle. We have to solve the dynamics of a charged Dirac fermion in the near-extremal near-horizon metric of the RN black hole, eq. (11). The logic of the computation follows the same steps explained in section 3 with extra technical complications due to the presence of the spin. The reader interested in the final result can directly jump to eq. (54). To solve the Dirac equation in curved space we use the vierbein formalism [44]. The Dirac equation is

$$\left[ \gamma^{(a)} e_{(a)}{}^{\rho} \left( \partial_\rho + \Gamma_\rho - ig A_\rho \right) + \mu \right] \Psi = 0 \ , \tag{34}$$

where $\mu$ is the mass and $g = qe$ the charge of $\Psi$. In four space-time dimensions the spinor $\Psi$ has four complex components. At each space-time point $x$ it is possible to define a locally inertial system of coordinates by introducing the tetrad $e^{(a)}{}_\alpha$ by means of

$$g_{\alpha\beta}(x) = e^{(a)}{}_\alpha(x) e^{(b)}{}_\beta(x) \eta_{ab} \ , \tag{35}$$

with Minkowski metric $\eta_{ab} = (-1, +1, +1, +1)$. Latin (greek) indices are raised and lowered with the Minkowski (general non-inertial) metric, $e_{(a)}{}^\alpha = e^{(b)}{}_\beta \eta_{ab} g^{\alpha\beta}$. Using this construction, we introduce generally covariant Dirac matrices $\gamma^\alpha$ defined as $\gamma^\alpha \equiv \gamma^{(a)} e_{(a)}{}^\alpha$, where the flat-space Dirac matrices $\gamma^{(a)}$ satisfy the usual relations $\gamma^{(a)} \gamma^{(b)} + \gamma^{(b)} \gamma^{(a)} = 2\eta^{ab}$. From the definition of the tetrad, we have $\gamma^\alpha \gamma^\beta + \gamma^\beta \gamma^\alpha = 2g^{\alpha\beta}$ that generalizes the algebra of the Dirac matrices in curved space. The spin connection $\Gamma_\rho$ entering in the definition of the covariant derivative in eq. (34) is given by

$$\Gamma_\rho = \frac{1}{8} \left[ \gamma^{(a)}, \gamma^{(b)} \right] g_{\mu\nu} e_{(a)}{}^\mu \nabla_\rho e_{(b)}{}^\nu \ , \tag{36}$$

where the covariant derivative acts on the tetrad according to $\nabla_\mu e_{(b)}{}^\alpha = \partial_\mu e_{(b)}{}^\nu + \Gamma^\alpha_{\mu\nu} e_{(b)}{}^\alpha$. We solve the Dirac equation in the so-called rotation frame defined by the tetrad

$$e^{(\tau)}{}_\tau = \frac{Q}{z} \left( 1 - \frac{z^2}{z_0^2} \right)^{1/2} \ , \qquad e^{(z)}{}_z = \frac{Q}{z} \left( 1 - \frac{z^2}{z_0^2} \right)^{-1/2} \ , \qquad e^{(\theta)}{}_\theta = Q \ , \qquad e^{(\varphi)}{}_\varphi = Q \sin\theta \ . \tag{37}$$

This choice lends itself particularly well to separate variables [45]. Eq. (34) becomes

$$\left\{ \gamma^{(0)} \frac{z}{\sqrt{f(z)}} \left[ \partial_\tau - \frac{igQ}{z} \left( 1 - \frac{z}{z_0} \right) \right] + \gamma^{(1)} z \sqrt{f(z)} \partial_z + \gamma^{(2)} \partial_\theta + \gamma^{(3)} \frac{1}{\sin\theta} \partial_\varphi + Q\mu \right\} \Psi = 0 \ , \tag{38}$$

where we used the short-hand notation $f(z) \equiv 1 - z^2/z_0^2$ and where we rescaled the spinor field according to $\Psi \to \Psi/\sqrt{\sin\theta}$. It is now possible to separate variables. The total angular momentum is $\vec{J} = \vec{L} + \vec{S}$. We combine spherical harmonics, which are eigenstates of $\vec{L}^2$ and $L_z$, and spinors, which are eigenstates of $\vec{S}^2$ and $S_z$, to form eigenstates of $\vec{J}^2$ and $J_z$. The latter are the so-called spherical spinors. The ansatz analogue to eq. (15) is

$$\Psi(\tau, z, \theta, \varphi) = e^{-i\omega\tau} \left[ Z_+(z)\Phi^+_{\kappa,m}(\theta, \varphi) + Z_-(z)\Phi^-_{\kappa,m}(\theta, \varphi) \right] , \tag{39}$$

where $\Phi^\pm_{\kappa,m}(\theta, \varphi)$ are spherical spinors in the rotation frame.[10] We obtain the radial equation

$$\left[ z\sqrt{f(z)}\partial_z \pm \kappa \right] Z_\pm(z) - i \left\{ \frac{z}{\sqrt{f(z)}} \left[ \omega + \frac{gQ}{z} \left( 1 - \frac{z}{z_0} \right) \right] \pm Q\mu \right\} Z_\mp(z) = 0 . \tag{43}$$

By introducing the combinations $\mathcal{Z}_\pm \equiv Z_+ \pm Z_-$, we find

$$\left\{ z\sqrt{f(z)}\partial_z \mp i \frac{z}{\sqrt{f(z)}} \left[ \omega + \frac{gQ}{z} \left( 1 - \frac{z}{z_0} \right) \right] \right\} \mathcal{Z}_\pm(z) + (\kappa \pm iQ\mu)\,\mathcal{Z}_\mp(z) = 0 . \tag{44}$$

These equations can be recast in matrix form by introducing the two-component vector $\tilde{\mathcal{Z}} \equiv (\tilde{\mathcal{Z}}_+, \tilde{\mathcal{Z}}_-)^{\mathrm{T}}$. We find

$$\left\{ \partial_z - \frac{i}{f(z)}\sigma_3 \left[ \omega + \frac{gQ}{z} \left( 1 - \frac{z}{z_0} \right) \right] \right\} \tilde{\mathcal{Z}} = \frac{Q}{z\sqrt{f(z)}} \left( \sigma_2\mu - \sigma_1\frac{\kappa}{Q} \right) \tilde{\mathcal{Z}} , \tag{45}$$

where $\sigma_{i=1,2,3}$ are the usual Pauli matrices acting on the two components of $\tilde{\mathcal{Z}}$. Let us pause for a moment to comment on this result. The ansatz in eq. (39) made possible the dimensional

---

[10] For spin $s = 1/2$ the possible values of $j$ are $j = l \pm 1/2$ for $l = 1, 2, \ldots$ and $j = 1/2$ if $l = 0$. As customary, we define the relativistic angular momentum quantum number $\kappa$ as

$$\kappa = \begin{cases} -l - 1 & j = l + 1/2 \\ l & j = l - 1/2 \end{cases} \implies \begin{cases} s & l = 0 & j = 1/2 & \kappa = -1 \\ p_{1/2} & l = 1 & j = 1/2 & \kappa = 1 \\ p_{3/2} & l = 1 & j = 3/2 & \kappa = -2 \\ \ldots \end{cases} \tag{40}$$

The values of $\kappa$ can be summarized as $\kappa = \mp(j + 1/2)$ for $j = l \pm 1/2$. The spherical spinors in the Cartesian frame $\bar{\Phi}^\pm_{\kappa,m}$ are defined by [45]

$$\bar{\Phi}^+_{\mp(j+1/2),m} = \begin{pmatrix} i\Omega^m_{j\mp1/2} \\ 0 \end{pmatrix} , \qquad \bar{\Phi}^-_{\mp(j+1/2),m} = \begin{pmatrix} 0 \\ \Omega^m_{j\pm1/2} \end{pmatrix} , \tag{41}$$

with

$$\Omega^m_{j-1/2} = \begin{pmatrix} \sqrt{\frac{j+m}{2j}}Y^{m-1/2}_{j-1/2} \\ \sqrt{\frac{j-m}{2j}}Y^{m+1/2}_{j-1/2} \end{pmatrix} , \qquad \Omega^m_{j+1/2} = \begin{pmatrix} \sqrt{\frac{j-m+1}{2j+2}}Y^{m-1/2}_{j+1/2} \\ -\sqrt{\frac{j+m+1}{2j+2}}Y^{m+1/2}_{j+1/2} \end{pmatrix} , \tag{42}$$

and $-l \leqslant m \leqslant l$. The spherical spinors in the rotation frame $\Phi^\pm_{\kappa,m}$ are related to $\bar{\Phi}^\pm_{\kappa,m}$ by means of a similarity transformation [45], and they satisfy the relations $\gamma^{(0)}\Phi^\pm_{\kappa,m} = \mp i\Phi^\pm_{\kappa,m}$, $\gamma^{(1)}\Phi^\pm_{\kappa,m} = \pm i\Phi^\mp_{\kappa,m}$, $\mathcal{K}\Phi^\pm_{\kappa,m} = i\kappa\Phi^\mp_{\kappa,m}$ with $\mathcal{K} = \gamma^{(2)}\partial_\theta + \gamma^{(3)}\frac{1}{\sin\theta}\partial_\varphi$.

reduction of a spinor $\Psi$ propagating in $AdS_2 \times S^2$ to a tower of spinor fields $\tilde{\mathcal{Z}}_\kappa$ in $AdS_2$, each one solution of eq. (45) for a fixed value of $\kappa$. In two dimensions, a Dirac spinor has indeed two complex components that are precisely $\tilde{\mathcal{Z}}_\pm$ introduced before. Eq. (45) can be decoupled and admit analytical solutions [46]. It is instructive to consider first the extremal case $z_0 \to \infty$ (that is the near-horizon limit of an extremal RN black hole). The general solution of eq. (45) in this limit is

$$
\begin{aligned}
\tilde{\mathcal{Z}}_\kappa(z,\omega) &= \frac{1}{\sqrt{z}} \left[ C_{\text{out}} \mathcal{W}_{-\frac{\sigma_3}{2}-igQ,\nu_\kappa}(2i\omega z) \begin{pmatrix} \tilde{\mu} \\ -1 \end{pmatrix} + C_{\text{in}} \mathcal{W}_{\frac{\sigma_3}{2}+igQ,\nu_\kappa}(-2i\omega z) \begin{pmatrix} -1 \\ \tilde{\mu}^* \end{pmatrix} \right] , \\
\nu_\kappa &\equiv \sqrt{Q^2(\mu^2-g^2)+\kappa^2} ,
\end{aligned} \tag{46}
$$

where $\sigma_3 = \pm 1$ when acting on the upper and lower component of the spinor and where we used the short-hand notation $\tilde{\mu} \equiv -\kappa/Q - i\mu$. Notice the close analogy with eq.s (18, 19). The ingoing boundary condition at the horizon selects the solution $\propto \mathcal{W}_{\sigma_3/2+igQ,\nu_\kappa}(-2i\omega\nu_\kappa)$. We now turn our attention to the $AdS_2$ boundary at $z \to 0$. In this limit eq. (45) takes the form

$$
z\left(\partial_z \tilde{\mathcal{Z}}\right) = U\tilde{\mathcal{Z}} , \quad U \equiv \begin{pmatrix} igQ & -iQ\mu-\kappa \\ iQ\mu-\kappa & -igQ \end{pmatrix} . \tag{47}
$$

The matrix $U$ can be diagonalized, and we find

$$
Uv_\pm = \pm\nu_\kappa v_\pm , \qquad v_\pm = \begin{pmatrix} \frac{1}{Q}(igQ \pm \nu_\kappa) \\ -\frac{\kappa}{Q}+i\mu \equiv \tilde{\mu}^* \end{pmatrix} . \tag{48}
$$

Using this result, it is simple to verify that eq. (47) is solved by

$$
\tilde{\mathcal{Z}}_\kappa(z,\omega) \stackrel{z\to 0}{\approx} \mathcal{A}_\kappa(\omega)v_- z^{-\nu_\kappa} + \mathcal{B}_\kappa(\omega)v_+ z^{\nu_\kappa} . \tag{49}
$$

Notice that in eq. (48) we fixed the arbitrary normalization of the eigenstates of $U$ to match the bottom component of the incoming solution in eq. (46). We can, therefore, easily extract (up to a $\omega$-independent normalization) the functions $\mathcal{A}_\kappa(\omega)$ and $\mathcal{B}_\kappa(\omega)$ directly from the asymptotic expansion of eq. (46) (obtained by means of eq. (20)). We find

$$
\mathcal{A}_\kappa(\omega) = \frac{\Gamma(2\nu_\kappa)}{\Gamma(1+\nu_\kappa-igQ)}(-2i\omega)^{1/2-\nu_\kappa} , \quad \mathcal{B}_\kappa(\omega) = \frac{\Gamma(-2\nu_\kappa)}{\Gamma(1-\nu_\kappa-igQ)}(-2i\omega)^{1/2+\nu_\kappa} . \tag{50}
$$

Using the definition of Green's function as the quotient of the sub-leading to leading term, we find

$$
\mathcal{G}_R^{(\kappa)}(\omega) = e^{-i\pi\nu_\kappa} \frac{\Gamma(-2\nu_\kappa)\Gamma(1+\nu_\kappa-igQ)}{\Gamma(2\nu_\kappa)\Gamma(1-\nu_\kappa-igQ)}(2\omega)^{2\nu_\kappa} . \tag{51}
$$

Notice the similarities with eq. (26).

According to the AdS/CFT dictionary, a bulk Dirac spinor $\tilde{\mathcal{Z}}_\kappa$ with charge $q$ is mapped to

a fermionic operator $\mathcal{O}_\kappa$ with the same charge in the CFT at the boundary.[11] The conformal dimension of $\mathcal{O}_\kappa$ is given by

$$\Delta_\kappa = \frac{1}{2} + \underbrace{\sqrt{Q^2 \left[ \mu^2 \left( 1 + \frac{\kappa^2}{\mu^2 Q^2} \right) - g^2 \right]}}_{\equiv \nu_\kappa \text{ in eq. } (46)} . \tag{52}$$

In analogy with the scalar case, there are no poles in the retarded Green's function in the extremal limit, and we need to investigate the near-extremal limit at non-zero temperature. The general solution of eq. (45) at finite temperature is a combination of hypergeometric functions (we do not report here the corresponding lengthy expressions). Using standard properties of the hypergeometric functions it is possible to identify the source term at the boundary and the quotient of the sub-leading to leading term. In close analogy with eq. (29), we find for the retarded Green's function (see also [46])

$$\mathcal{G}_R^{(\kappa)}(\omega) = (4\pi T)^{2\nu_\kappa} \frac{\Gamma(-2\nu_\kappa)\Gamma(1/2 + \nu_\kappa - i\omega/2\pi T + igQ)\Gamma(1 + \nu_\kappa - igQ)}{\Gamma(2\nu_\kappa)\Gamma(1/2 - \nu_\kappa - i\omega/2\pi T + igQ)\Gamma(1 - \nu_\kappa - igQ)} . \tag{53}$$

The quasi-normal frequencies are dictated by the poles of the Gamma function, and for the imaginary part we find

$$\mathbb{Im}[\omega_{n,\kappa}] = -2\pi T \left( \frac{1}{2} + n + \mathbb{Re}[\nu_\kappa] \right)_{n=0,1,\dots} , \qquad \nu_\kappa = \sqrt{Q^2 \left\{ \mu^2 \left[ 1 + \frac{\kappa^2}{\mu^2 Q^2} \right] - g^2 \right\}} \tag{54}$$

This result is remarkably similar to eq. (30). Following the discussion in section 3 we concentrate on the lowest mode with $n = 0$ and $l = 0$. Because of the non-zero spin, we have $\kappa = -1$, as discussed in eq. (40). The thermalization time-scale is given by

$$\tau_d \equiv \tau_d^{(0,\kappa=-1)} = \frac{1}{T(1/2 + \nu_{\kappa=-1})} , \qquad \nu_{\kappa=-1} = \sqrt{1 + Q^2 (\mu^2 - g^2)} \tag{55}$$

The condition $\nu_{\kappa=-1} < 1/2$ guarantees the condition $\tau_d > 1/T$ on the thermalization time-scale and, after restoring units of $M_{\mathrm{P}}$, we find

$$qe > \frac{\mu}{M_{\mathrm{P}}} \tag{56}$$

that is the WGC.

We conclude that the same proof of the WGC put forward in section 3 for a charged scalar particle remains true also in the case of a Dirac fermion.

---

[11] It is instructive to count components of fermions. In dimensions $d = 2n$ (even) and $d = 2n + 1$ (odd) a Dirac spinor has $2^n$ complex components and $2^{n-1}$ complex degrees of freedom. Thus in $d = 4$ ($n = 2$) a Dirac fermion has 4 complex components and 2 complex degrees of freedom (spin up and spin down for the particle, and further two for the anti-particle). In the $AdS_2$ bulk we have $d = 2$ ($n = 1$) and a Dirac fermion has 2 complex components and 1 complex degree of freedom. The mismatch between components and degrees of freedom is due to the first order nature of the Dirac action. From the Dirac Lagrangian, it follows that the momentum conjugate to the spinor $\psi$ is given by $\pi_\psi = i\psi^\dagger$. The phase space of the Dirac fermion has $2 \times 2^n$ real dimensions and correspondingly the number of real degrees of freedom is $2^n = 2 \times 2^{n-1}$. In the context of the $AdS_2/CFT_1$ correspondence, it means that only half of the components of $\tilde{\mathcal{Z}}_\kappa$ correspond to a boundary fermionic operator $\mathcal{O}_\kappa$.

# 5 Discussion and conclusions

The WGC asserts a powerful consistency condition on gauge theories coupled to gravity. However, little is known about the physics from which it originates. In this paper we investigated the WGC from a thermodynamic perspective, and the main motivation to pursue this route can be summarized as follows.

The original arguments that motivated the WGC were based on black hole physics, and the basic properties of black holes can be expressed as a fairly simple set of rules known as black hole thermodynamics. More ambitiously, the connection between gravity and thermodynamics lies at the heart of the AdS/CFT correspondence in which the thermodynamics of black holes is identified with thermodynamics of CFTs. All these arguments bring out the idea that gravity and thermodynamics are intimately linked.

It is natural to ask whether the WGC admits a thermodynamic formulation. In this paper, a positive answer is given: *the WGC is related to the thermalization dynamics governing the relaxation process after a perturbation, and its validity is guaranteed by the existence of a lower bound on the thermalization time-scale.* The latter is known as the universal relaxation bound. This thermodynamic interpretation of the WGC was already proposed in [23]. The novelty we added in this paper is that we came to the same conclusion by studying the dynamics of test fields – both scalar and fermions – in the near-extremal near-horizon geometry of RN black hole. This limit is particularly interesting since the isometry group of the black hole is enhanced to $AdS_2 \times S^2$ thus allowing, after KK reduction on $S^2$, the formulation of the dynamics in terms of an AdS/CFT-type correspondence. This is interesting since CFTs duals to RN black holes admit a lower bound on the thermalization time-scale of the same form of the one used to prove the WGC, thus corroborating the validity of the positivity constraints used in section 3, and providing a dual description for the universal relaxation bound.

A final comment about the AdS/CFT correspondence is in order. First of all, it is more correct to talk about a "near-AdS/near-CFT" (nAdS/nCFT) correspondence. In our construction, this is because the $AdS_2$ metric is obtained by carrying out a KK reduction over the sphere $S^2$. The latter has compactification radius equal to the black hole charge $Q$. Consequently, $1/Q$ characterizes the scale at which corrections to the AdS geometry become significant, and the pure AdS/CFT limit only applies at low energy and temperature. We did not explore in depth the duality in this paper (for related studies, cf. [47,48]) but it is worth emphasizing that it shares remarkable similarities with the SYK model of strange metals. We stress once again that the SYK model admits a lower bound on the thermalization time-scale of the same form of the one used to prove the WGC.

The SYK model realizes a $nAdS_2/nCFT_1$ correspondence between a one-dimensional quantum mechanical system consisting of $N$ Majorana fermions with random four-fermion interactions (the nCFT side of the correspondence) and a two-dimensional dilaton-gravity theory with a negative cosmological constant first studied by Jackiw and Teitelboim (JT) [49,50] (the nAdS side of the correspondence). In the SYK model the attribute near- is due to the fact that the conformal symmetry consisting of all time-reparametrization is both spontaneously

and explicitly broken. On the gravity side, the $AdS_2$ geometry is broken by a non-constant dilaton field [51].

The connection with the setup studied in this paper is due to the fact that the near-horizon geometry of near-extremal RN black holes shares some properties with the JT model. To be more specific, the JT theory describes the s-wave sector ($l = 0$) of higher dimensional theories featuring an $AdS_2$ throat geometry after compactification (e.g., cf. [52]). This is the case of the reduction from $AdS_2 \times S^2$ to $AdS_2$ exploited in this paper. In our setup, the role of the dilaton is played by the radius of the sphere $S^2$, and a non-constant profile can be obtained by introducing small fluctuations around it [47, 48]. In light of the computation proposed in this paper, it is interesting to explore more the connection with the JT model. We hope to clarify some of these aspects in the near future. Assuming our speculations are correct, a proper identification of the nAdS/nCFT correspondence valid for near-extremal RN black holes will put on more solid ground the universal relaxation bound $\tau_d > c/T$ that is crucial for the proof of the WGC.

# Acknowledgments

I wish to thank Subir Sachdev for useful comments.

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
