# Peer review of "Towards a proof of the Weak Gravity Conjecture"

_SciPost Physics_

## Round 2 · Referee Report · Anonymous · 2018-12-29

Strengths

1. This paper contains calculations of the quasinormal modes of charged scalars and fermions in the near-horizon background of an extremal Reissner-Nordstrom black hole.

2. It suggests a novel approach toward proving the Weak Gravity Conjecture, namely that a bound on the imaginary part of the quasinormal mode frequencies may follow from CFT properties using the AdS$_2$/CFT$_1$ correspondence.

Weaknesses

1. The paper is rather poorly organized. Neither the abstract nor the introduction explains the logic of the paper. It is only by reading the entire paper that the reader can understand what is accomplished.

2. The paper's abstract promises more than it actually achieves. It implies that the WGC is related to thermalization properties of CFTs and to the SYK model. However, the paper contains only very sketchy gestures in these directions, rather than new results.

3. Related to both of the above: the main arguments of the paper are closely related to previous arguments of Shahar Hod (ref [23] of the paper). The novelties relative to Hod's paper are, first, the use of the near-horizon limit to simplify the arguments; second, the treatment of charged fermions rather than just scalars. However, ref [23] is only cited briefly in passing in a couple of places, so a casual reader is likely to think that this paper is more original than it is.

4. The summary of previous literature in the introduction is, in my opinion, rather misleading. Furthermore, the referencing throughout the paper is poor.

Report

Most of my comments have already been summarized under "strengths" and "weaknesses" above. To provide more context:

An intriguing 2017 preprint by Shahar Hod claimed to "prove" the Weak Gravity Conjecture via an argument about black hole quasinormal mode frequencies. Hod claims that there is a "universal relaxation bound" on quantum systems which requires that a mode exist with a sufficiently small imaginary part of its frequency. In other words, consistent quantum systems cannot relax too quickly. Hod observes that although Schwarzschild black holes obey such a bound, nearly-extremal Reissner-Nordstrom black holes do not: the imaginary parts of the frequencies of both graviton and photon quasinormal modes are too large. However, he argues that if a sufficiently light charged scalar exists, the bound will be obeyed. Furthermore, the condition for "sufficiently light" looks precisely like the WGC bound. Hence (he claims), obeying the universal relaxation bound requires that light charged fields exist. (A more accurate statement would be that the existence of light charged scalar fields provides one way to satisfy the bound; an argument that there are no other ways to satisfy the bound has not, to the best of my knowledge, been given, either by Hod or in the current paper.)

From my point of view, the "universal relaxation bound" itself is not a well-established statement about general theories of quantum gravity, and so any progress on establishing it from first principles would be important.

Hod's argument is intriguing, and so far very little work has been done to follow up on it. Hence, I think the current paper, which takes two interesting steps toward following up (first, to show that the near-horizon limit can streamline the arguments and derivations; second, to consider charged fermions), is a useful contribution to the literature. However, the way that this paper is written obscures the link to Hod's earlier paper. Furthermore, the introductory sections offer vague claims about linking the WGC to thermalization properties, but we only encounter the actual argument after reading through the detailed derivations in sections 3 and 4. I could imagine that more rigorous arguments about CFTs might help to make Hod's "universal relaxation bound" more convincing, but such arguments are not given in this paper. The paper waves in the direction of SYK models, but makes no really concrete connection to them, which leaves the reader with the feeling that this has been dropped in just because it is a popular buzzword.

My expectation is that the actual computations in the paper are correct, though I have not checked them in detail. The results are plausible (and in some cases can be checked against claims in earlier literature). I have focused my attention on the bigger structural problems of the paper.

Requested changes

1. The paper should be thoroughly restructured so that readers understand, from reading the abstract and introduction, what it accomplishes that is novel. It should not over-promise results that it does not deliver. The calculations are an interesting refinement of Hod's arguments, and I think that that is sufficient for the paper to be interesting and publishable.

2. References to the earlier literature seem to have been chosen almost at random. For instance, the statement (equation 9) that the near-horizon geometry of extremal Reissner-Nordstrom is ${\rm AdS}_2 \times {\rm S}^2$ has been well-known for many years before the Guica et al. paper on the Kerr/CFT correspondence (cited as ref [27]). The author should make an effort to cite the literature more appropriately throughout the paper.

3. The author gives some rough arguments that CFTs should obey a bound $\tau > c/T$ for some order-one coefficient $c$. It should be noted that the WGC is obtained only for a particular value of this coefficient. Hod gave an argument for what this coefficient should be, but the vaguer arguments suggested in this paper seem to only produce a WGC-like result rather than a result with a sharp coefficient. The author should clarify this; e.g. should equation (56) contain a factor of $c$? Or has the coefficient been fixed by reference to Hod's claimed relaxation bound?

4. In my opinion the introduction is rather misleading regarding the status of the WGC. The older "trouble for remnants" arguments really apply to remnants that all lie at the same mass scale and do not apply to the charged black holes of varying mass that arise if the WGC is violated. In particular, there are consistent theories with infinite families of stable BPS black holes of varying charge and mass, so it does not seem that such arguments could really prove the WGC. Similarly, the "infrared consistency" arguments of ref [5] do not prove anything resembling the WGC, since they depend on unjustified assumptions about the coefficients of higher-derivative operators at the string scale. So in my opinion #1 and #3 are strongly over-stated. Statement #2, "No global symmetries," is summarized more accurately, but referenced poorly: these arguments predate the Banks & Seiberg paper that is cited by decades, and earlier literature should be cited. Finally, statement #4, "Absence of counter-examples," understates the evidence in string theory, which goes far beyond "a handful of examples." In this context ref [9] ("Evidence for a sublattice weak gravity conjecture") as well as Montero et al.'s "The Weak Gravity Conjecture in three dimensions" (arXiv:1606.08438) have shown that modular invariance implies (a stronger version of) the WGC in infinite families of string theory examples. Grimm et al.'s "Infinite Distances in Field Space and Massless Towers of States" (arXiv:1802.08264) shows that the WGC is satisfied by towers of BPS states in yet another huge class of string theory examples. These papers should be cited alongside reference [6], and the Montero et al. paper should also be cited later where refs [8-10] are cited.

---

## Round 2 · Referee Report · Anonymous · 2019-1-21

Strengths

- New arguments based on the near-horizon limit of RN black holes.
- Qualitative use of AdS/CFT to rephrase the weak-gravity conjecture as
thermalization property of the dual field theory

Weaknesses

- The paper is largely based on a previous papers (reference [23]) by S. Hod.
- The validity use of the near-horizon geometry in all arguments is not completely obvious to me, when it comes to the asymptotic behaviour and regularity of the solution.
- The numerical coefficient leading to eq. (33) seems to have been chosen in an ad-hoc way.
- I don't see any explicit connexion to the SYK model.

Report

The paper, largely base on [23], gives some additional arguments towards a proof of the weak-gravity conjecture (WGC), based on the near-horizon geometry, relating the WGC condition, eq. (33), to the thermalization properties of a conjectured dual field-theory. The paper gives additional arguments, based on the propagation of scalar and fermionic perturbations in the RN near-horizon geometry, that support the claim that the known limits on the thermalization time-scale maps into the WGC condition, eq. (33).
The paper is an interesting follow-up of [23] that contain new contributions that deserve publication.
I propose however some clarifications in the text prior to publication, see below.

Requested changes

- The reference [23] should be given more credit. In particular, it should be made clear
that the main purpose of the paper is to substantiate the results of [23] by using
additional arguments.
- It is not clear to me that the numerical coefficient in the thermalization time limit of the dual field-theory is precisely the one needed for the WGC condition, eq. (33).
Unless I miss this point, the paper should state clearly that this is an open question.
- Although the paper gives some arguments for why the near-geometry is enough for the study of low-frequency perturbations at low-temperatures, it is not clear to me
that conditions on the regularity of the perturbations is compatible with the near-horizon limit. Some further comments on this point would be useful.

---

## Editorial Decision

awaiting_resubmission